# The Construction of the Visible and Invisible Boundaries of Microsegregation: A Case Study from Szeged, Hungary

Ramóna Vámos [1], Gyula Nagy [1] and Zoltán Kovács [1,2,*]

1   Department of Economic and Social Geography, University of Szeged, 6720 Szeged, Hungary;
    vamos.ramona@geo.u-szeged.hu (R.V.); gynagy@geo.u-szeged.hu (G.N.)
2   Geographical Institute, HUN-REN Research Centre for Astronomy and Earth Sciences,
    1112 Budapest, Hungary
*   Correspondence: zkovacs@iif.hu

**Abstract:** The concept of microsegregation has gained increasing popularity among researchers dealing with socio-spatial disparities in cities. This is because urban space has become increasingly multifaceted over recent decades, and the boundaries of socio-spatial segregation have also become increasingly subtle, often taking invisible forms below the neighborhood level. This study contributes to the literature on microsegregation by exploring small-scale forms of social disparities in one of the neighborhoods of Szeged, a second-tier city in Hungary. We used both quantitative and qualitative research methods to capture visible and invisible forms of microsegregation in the study area. An analysis of census data confirmed the coupling of socio-economic diversity and polarization at the census-tract level in three different forms as a result of various underlying factors, among which the sorting effect of the housing market plays a leading role. The results of in-depth interviews with experts and residents suggest that although the overall perception of the neighborhood is good and that serious conflicts do not occur, there are palpable socio-spatial differences and signs of segregation at the micro scale. The weak sense of segregation can be partly linked to the lack of public spaces where daily encounters between people from different social groups could take place.

**Keywords:** residential segregation; microsegregation; social mix; intersectionality; diversity; mapping microsegregation; post-socialist city

## 1. Introduction

The seminal work of Burgess (1923), published 100 years ago, not only laid the foundations of human ecology but also focused the attention of researchers on urban segregation [1,2]. The city of Chicago, where the first urban social theory was developed, had large and noticeable segregated spaces [3]; therefore, early observations on segregation referred to large-scale horizontal spaces with distinct ethnocultural variations. Ever since, new approaches have appeared in the study of segregation, ranging from behavioral to structural theories, and, most recently, from examining the welfare state to political theory, with emphasis on different underlying forces of segregation [4]. However, the proliferation of theoretical approaches has, to some extent, overshadowed the relevance of the geographical scale. Segregation is a multiscale process that should not be overlooked by researchers.

Segregation in cities has been discussed in the literature, mainly in the context of racial, ethnic (religious), and socio-economic (class) characteristics [5–8]. Such forms of segregation usually manifest at visible boundaries on the macro scale (i.e., neighborhoods, districts, and larger urban zones), and they are reaffirmed via distinct features of the built environment, such as certain roads, fencing, gating, or landmarks of different cultures (e.g., churches). However, the social geography of cities has changed tremendously over the past few decades, which may be linked to the increasing professionalization of the urban workforce, intensification of migration, the mixing of different groups of people, rising levels of income

inequality, and—at least in the countries of the European Union—systematic policies aimed at enhancing social and ethnic mixing within urban neighborhoods. As a consequence of recent urban regeneration programs and multifaceted urban development, higher-income households tend to informally colonize neighborhoods that were previously considered less attractive and inhabited mostly by lower-class people. This means that statistically, social mixing increased in neighborhoods [9]; nevertheless, the coexistence of the rich and poor strata in certain areas of cities does not necessarily mean that daily interactions occur between residents, or that a cohesive local society exists. Conventional approaches to segregation fail to capture those situations where parallel societies coexist; therefore, in highly diversified neighborhoods, the boundaries delineating segregation often remain invisible [10]. To unpack such microscale differences in urban space, social geographers need to adopt new theoretical and methodological approaches.

The concept of microsegregation refers to urban milieus below the neighborhood level, where individuals living in close spatial proximity occupy unequal positions according to their socio-economic status or ethno-racial identity [11,12]. Contemporary urban processes, such as gentrification, the vertical expansion of cities, the spread of gated communities, and systematic policies to enhance social or ethno-racial mixing in neighborhoods, provide numerous opportunities for various forms of spatialized social hierarchy on a micro scale. Even though this type of segregation is not a primary issue for current urban policies, it is a persistent form of social hierarchy that affects the everyday life of most of the world's urban population. Another concept that deserves attention is intersectionality; this term has been used by scholars to describe an analytical framework of social justice [13–15]. Intersectionality has recently been employed in research focusing on issues relating to race, age, gender, and social class as it exists in urban space; hence, it is a useful tool for mapping and understanding both individual and intersectional structures of inequality and power.

The main objective of this study is to add to the literature on microsegregation by studying microscale forms of social disparities in one of the neighborhoods of Szeged, a second-tier city in Hungary. In this study, we used both quantitative and qualitative research methods to capture visible and invisible forms of microsegregation in our case-study area, and we attempted to explain the underlying factors of small-scale inequalities.

In the remainder of the study, first, we discuss the most relevant literature on microsegregation, followed by the introduction of the study area, a discussion of our research methods, and the main results of the research. Finally, we discuss the main findings and conclude our study.

## 2. Materials and Methods

Even though the concept of microsegregation has only recently become popular in the literature, the phenomenon was observed and documented much earlier in major European cities such as Naples and Paris [16,17]. Over the past 30 years, the relevance of the concept has increased in urban social theory as research on various aspects of microsegregation has intensified, and an increasing number of cities (e.g., Athens, Shanghai, and Tel Aviv) have become subjects of investigation [18–20]. Despite the growing number of studies that aim to capture and understand small-scale social segregation, there is still an immense gap regarding its definition, forms, and possible conceptual and methodological approaches.

The first studies on microsegregation focused on vertical residential segregation in Athens [21,22]. These were followed by more recent studies in Bucharest [23] and Marseille [24]. Vertical segregation is one of the most classic forms of microsegregation; however, it is not the only possible manifestation of the phenomenon that is worth studying. As the topic has gained growing attention, researchers have started to explore other types of microsegregation, such as horizontal segregation below the neighborhood level and small-scale gentrification [25–27]. The authors have concluded that the spatial manifestation of microsegregation depends very much on local specificities, such as social stratification or the role of the state in the housing market [23,24]. According to previous research results, microsegregation can be seen as the outcome of the interplay of the conscious decisions of

people at the micro scale and the framework conditions, primarily housing policy, sustained by the state at the macro scale. These factors change over time (e.g., due to globalization and migration) and may give rise to new forms of microsegregation. In this respect, following Simandan's recent analytical work [28], our study traces the transformation of space and society on a geographical scale, juxtaposing personal change and wider societal change.

Research on (micro)segregation has revealed most often the ethnic/racial/cultural aspects of the phenomenon; however, very little has been said about its socio-economic aspects (e.g., occupation and income). Recent research on small-scale segregation in schools has focused on exploring differences on a micro-geographical scale, and results suggest that family background is a basic factor of children's segregation within schools and that intermixing families with different socio-economic backgrounds does not necessarily result in good neighbor relationships between them [29–33].

Numerous studies suggest that microsegregation, described in different countries and cities, can manifest in a great variety of ways, and the main disadvantage of this diversity is that no single, commonly accepted definition has been developed yet. A key aspect of the most commonly used definition of microsegregation is scale, where the upper limit of the study area(s) tends to be at the neighborhood level, but small-scale census tracts are also often studied, while the smallest units of analysis are at the building/block level. Most recently, in their study on Hong Kong, Ho and Yip [34] introduced the concept of nanosegregation, which refers to the coexistence of multiple households with different socio-economic statuses in a single apartment. Despite these recent developments, we believe that the importance of understanding the neighborhood should not be overlooked in segregation studies. Neighborhoods as "lived-in" spaces provide an opportunity for geographers to study the everyday interactions of residents, their lived experiences of the neighborhood, and concomitant spatial formations [35]. The spatial formation of the neighborhood is multiscalar, which provides geographers with an opportunity to study who lives in a neighborhood and how invisible boundaries of residential segregation are created [36]. Another challenge in the study of microsegregation is that even though the process shares many similarities with the large-scale horizontal segregation of neighborhoods, the methodology of the latter is not fully applicable because more attention needs to be paid to small-scale dimensions of space (e.g., verticality, streets, and blocks). As a result of microscale mechanisms, segregation and social isolation within buildings, streets, and blocks become detectable, and urban space becomes much more variegated than the previous observations by researchers in the context of spatial segregation at the macro-urban horizontal level.

A common feature of previous studies on microsegregation is that they have been carried out in major cities at the top of the urban hierarchy, focusing mostly on the ethnic aspects of segregation, and particularly on vertical residential segregation in the skyscrapers of US and Asian cities [19,37]. Both the ethnic/religious aspects and verticality are present in studies focusing on microsegregation in European cities (see e.g., [8]); however, the majority of studies lag behind the US and especially Asia. This can be explained by the historical pathway of European urban development, long-term urban traditions, and stricter regulations regarding building heights and verticality. However, European countries are not homogeneous either; the development of the former state-socialist Central and Eastern European countries diverged from the Western pattern of urbanism after World War II (e.g., the dominance of the state in the housing market). State socialism could be characterized by high levels of employment, homogenization, and egalitarianism in the labor and housing markets as the main principles of societal life [10]. This system was disrupted abruptly in 1989/90 with the collapse of state socialism. In the early 1990s, the dismantling of central planning and the shift to a market economy resulted in a widening gap in income distribution within these societies, which was intertwined with social differentiation of urban space, including the upgrading and downgrading of neighborhoods, gentrification, and ghetto-formation, and the emergence of poverty and wealth [38,39].

All these processes have been present in Hungary since the early 1990s, which turned the attention of researchers to urban social segregation and resulted in a growing number of investigations. The marginalization of low-income groups, including the Roma population, and their distinct presence in the cityscape, has been the subject of several studies [40–43]. Research on segregation in Hungary has not only covered the fields mentioned above but has also focused on segregation at work and in schools. The concept of microsegregation has not been neglected in Hungary either, with studies focusing on Budapest (and comparing it with Athens and Bucharest) [44–46]. However, these studies deal exclusively with Budapest, the capital city of Hungary. The main advantage of cities at the top of the urban hierarchy is that they are not only destinations for domestic but also international (global) migration, providing various opportunities for distinct (visible) forms of segregation. The size and diverse societies of global cities make them popular locations regarding (micro)segregation research, whereas research on segregation, especially on the micro forms of the phenomenon, in cities at lower levels of urban hierarchy, is largely missing [47].

Szeged is the third most populous city in Hungary, situated on the southern periphery of the country [48], and it is one of the rare examples where the process of ethnic segregation has been studied intensely in recent years [41,49–51]. Publications in the field reflect the fact that segregation of different socio-economic and ethnic groups is an equally important issue at the lower levels of urban hierarchy. In the early 2010s, research on school (de)segregation in Szeged addressed the conditions of pupils in the school system in depth [52]. Several studies have been published examining segregation in Szeged; however, the phenomenon of microsegregation has not yet been addressed.

Therefore, the main objective of this study is to address the aforementioned gaps, providing evidence about different forms of microsegregation in a relatively small neighborhood at the lower level of urban hierarchy.

## 3. Research Design

### 3.1. The Study Area

The study area, Rókus, is one of the neighborhoods in Szeged, the third largest city of Hungary, located in the southeastern part of the country adjacent to the border with Serbia (Figure 1). According to the Hungarian Central Statistical Office (HCSO), the city had a total population of 168,000 on the eve of the 2011 census, and it is the primary administrative, economic, and cultural center of southeastern Hungary [53].

The study area is the home of approximately 9300 inhabitants (2011), i.e., about 5.5% of the city's population. In terms of the age and physical parameters of the building stock and the socio-demographic characteristics of the residents, the neighborhood is highly diversified. The inner part of the quarter, close to the city center, has a traditional, small-town character, with apartment buildings of two–three stories, mixed with terraced houses and semi-detached houses built predominantly at the turn of the 20th century. Architecturally, the buildings follow a simpler version of eclectic and Art Nouveau styles. Its original population came from traditional farms in the surrounding area, and due to widespread pig breeding, the area was called "Kukoricaváros" (Corn City), and its inhabitants were called "Kukoricapolgár" (Corn Citizens). By the beginning of the 20th century, the area became dominated by lower-rank officials, artisans, soap makers, and pig farmers. After World War II, the neighborhood was affected by rapid transformation and upgrading. As part of the state-socialist housing policy, some of the traditional, single-family houses were demolished in the northwestern part of the neighborhood and were replaced by multistory, prefabricated residential buildings as part of a large housing estate. The 1970s and 80s brought about further densification and transformation of the neighborhood with new public (e.g., schools, kindergartens) and commercial functions, and, consequently, its rural appearance gradually changed into a high-density and high-rise urban landscape [54].

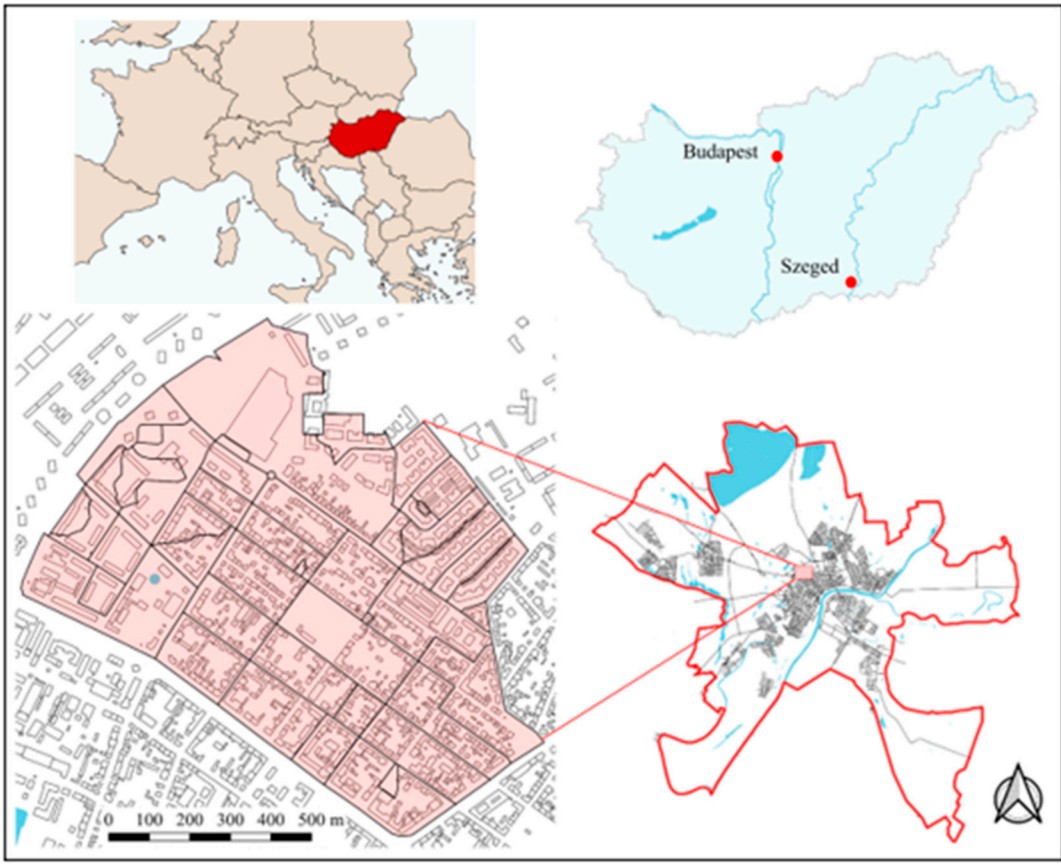

**Figure 1.** The study area: the Rókus neighborhood in Szeged, Hungary.

The transformation process of the traditional Rókus neighborhood continued even after the collapse of state socialism in 1990. The old housing stock deteriorated rapidly, and some of this was replaced by newly built, high-quality condominiums after the turn of the millennium. The latest housing development in the area is the "Franciahögy" residential park, an upmarket gated community built on the former site of the Rókus cemetery. The new infills and the gated community resulted in an influx of upper-middle-class families to the area and gave impetus to the rise of its prestige. All these developments resulted in the rapid change in the building stock of the neighborhood and the transformation of its society. Today, the neighborhood can be considered one of the most diverse areas of Szeged with the potential to manifest various forms of microsegregation (based on age, education, ethnicity, socio-economic status, etc.). However, due to its complexity, the segregation of the area cannot be interpreted in the traditional sense of neighborhood-scale segregation; instead, it is manifested at the lower geographical scale, at the block and street level.

### 3.2. Data and Methods

This study approaches microsegregation from a new methodological perspective, employing a mixed-methods approach. During the study, we used both quantitative and qualitative research methods to unpack different forms of microsegregation in the neighborhood of Szeged.

As part of the quantitative analysis, the housing and socio-economic data of residents were obtained at the census-tract level from the Hungarian national census of 2011. The last census of Hungary was held in November 2022 (due to a postponement caused by COVID-19 in 2021), but data with fine spatial resolution are not yet available. However, we think that the 2011 census data, collected by the HCSO, are sufficient to capture microscale socio-economic differences within the neighborhood under investigation. The focus of the study was on the positions of residents in the occupational structure and their demographic

and household characteristics. Notably, we used the International Standard Classification of Occupations 2008 (ISCO08) to group active earners according to three broad categories, namely Managers and Professionals (higher ISCO08 categories 1 and 2); Technicians, Office Employees, and Service Workers (intermediate ISCO08 categories 3, 4, and 5); and working-class categories comprising manual laborers in industry, construction, transport, etc. (lower ISCO08 categories 6, 7, 8, and 9). These three ISCO08 categories roughly reflect the socio-economic status of residents. In addition to occupational data, we used other variables (e.g., demographics, size, and comfort of housing) from the census to depict microscale boundaries of segregation in the study area.

To examine the internal heterogeneity of the study area, we used the Shannon–Weiner Species Diversity Index for the ISCO data at the census-tract level. The diversity index is widely used in biology, and it is a quantitative measure that reflects how many different types of species there are in a community. It is calculated by taking the number of each species, the proportion of each species of the total number of individuals, and summing the proportion times the natural log of the proportion for each species.

$$H = -\sum_{i=1}^{s} \ln p_i * p_i$$

where $H$ is the species diversity index, $s$ is the number of species, and $p_i$ is the proportion of individuals of each species belonging to the $i^{th}$ species of the total number of individuals. Higher values reflect higher diversity in the area; the lower the value, the more statistically probable the dominance of one or two social groups. Diversity and polarization should be interpreted in the following way. Although diversity refers to a heterogeneous social environment that contains various social groups, polarization means that heterogeneity is the outcome of the presence of two opposing groups in the area, i.e., in this case, the top and bottom occupational groups.

To detect diversity and polarization simultaneously, we also took into consideration if the diversity of a census tract is caused by the presence of the highest and lowest ISCO categories together. To capture polarization within diversity, a three-fold legend was created labeling the low, average, and high values. The higher the H-index value, the more diverse the census tract in terms of the different ISCO groups. The lower the H-index value, the higher the probability of the presence of a dominant ISCO group; therefore, a lower *H*-value can also mean a more polarized census tract. However, we were interested in polarization, which was caused by the simultaneous presence of both the low and high ISCO groups. This also meant that the dominance of two specific ISCO groups could result in lower diversity values in some of the census tracts.

To determine when diversity and polarization coexist, we considered the share of the two opposite ISCO groups (ISCO 1, 2 and ISCO 7, 8, 9 occupational groups), and we compared their distribution to a perfectly equal distribution (where each group has the same number of individuals). Since the upper-status occupational group represents two categories and the lower group represents three categories, we looked for cases where the share of ISCO 1 and 2 exceeded 20% and ISCO 7, 8, and 9 exceeded 30%. Only two census tracts met this strict condition. Subsequently, we examined whether the values within each group exceeded the average values measured in all census tracts. This allowed us to identify tracts that are not only diverse but also show both ISCO groups to be above or very close to the average value. Thus, we established the final delineation of the census tracts.

The quantitative analysis was elaborated by qualitative data collection. This phase of the research was based on semistructured in-depth interviews with experts and residents. First, we conducted structured interviews with experts who are familiar with the area and have insight into its social processes. The interviewees included teachers, school psychologists, and real estate agents. Second, we interviewed residents of the neighborhood who experienced its social and community processes from the inside. Altogether, 16 interviews were conducted between March and June 2023. The interview-based research had three main focal points.

The primary focus was on the spatiality of social segregation. At the beginning of the interview, we asked respondents whether they observed any form of social segregation in the neighborhood and whether there were noticeable spatial boundaries as regards the socio-economic status and attitude and lifestyle of the residents. This question aimed to validate the quantitative research findings with qualitative data. Additional questions also focused on mental boundaries and whether any perceived boundaries were caused by experiencing fear or harassment in the area. Concerning segregation and dividing fault lines in the sample area, we were also interested in whether respondents experienced or heard about any conflicts between different social or ethnic groups, and if there were any visual (graffiti or wall writings) or spatial (avoided streets) manifestations of such conflicts. The final set of questions focused on the general socio-economic characteristics and lifestyle of the interviewees. This aimed to determine whether there was any difference in terms of the use of space, everyday activities, and use of services. Based on the answers, we tried to define the invisible boundaries of social segregation and their construction in the study area.

We were also interested in whether the delineation of census tracts may hide certain forms of microsegregation at the street or block level from a statistical perspective, making them invisible simply by drawing boundaries. Therefore, we paid special attention to those areas where, based on previous experiences and a preliminary field survey, the occurrence of microsegregation was adjudged possible. We treated these areas as a priority in the qualitative survey to gather evidence of boundary drawing that might blur microsegregation.

## 4. Results

### 4.1. Quantitative Aspects of Segregation in Rókus

Population and housing statistics from the 2011 National Census reflect high diversity in the study area of Rókus within the city. The demographic structure of the population is younger, and the share of active earners is significantly higher than the city's average (Table 1). Taking into account the level of education, the ratio of people with tertiary education is also well above the city's average and consequently, the share of the top socio-occupational categories (ISCO1–2: Managers and Professionals) is also significantly above the overall value of Szeged. Conversely, the share of lower socio-economic groups (ISCO7–9: industrial workers, machine operators, unskilled workers) does not deviate much from the city's average. This implies higher-than-average socio-economic diversity and polarization within Rókus. Polarization is also confirmed by data on the local housing market. The share of old housing (built before World War II) is similar to the average of the city; however, the share of the new housing stock is much higher. More than half of the local housing stock was built after the political changes of 1990. The higher share of new housing indicates that the ratio of smaller (one-bedroom) dwellings is nearly double the city's average. We can conclude that basic statistics from the last census reveal a high level of mix regarding both the socio-economic profile of residents and the housing stock in the area. Now, it is an intriguing question how unevenly different housing forms and socio-economic groups are distributed in the area and what types of microscale inequalities can be detected in Rókus.

As Figure 2 shows, old, pre-socialist housing stock (in multistory apartment buildings built before 1946) prevails in the southeastern part of the neighborhood. This part of the study area also contains some public rental units, which are scattered throughout the census tracts and are of generally low quality (see (1) of Figure 3). Post-World War II socialist-era housing in the form of prefabricated 4- and 10-storey blocks occupies the northwestern part of the neighborhood with standardized architecture and homogeneous quality [55] (see (2) of Figure 3). This part of the neighborhood was reconstructed between 1978 and 1990 in three phases when old low-quality and low-rise buildings were demolished and replaced by high-rise housing built by the state with prefabricated panel technology. The level of comfort rose, uniform two-room apartments with bathrooms and central (district) heating became dominant, and consequently, the living conditions of residents also improved

significantly. As some of the literature has noted in the housing estates of the 1970s and 80s, the proportion of less-educated working-class people was higher among the new residents compared to the previous decades when the allocation of new state-built housing was more elitist [56,57]. Therefore, we expect above-average social mixing in this part of the local housing stock.

**Table 1.** Population and housing characteristics of Szeged and Rókus.

|  | **Szeged** | **%** | **Rókus** | **(%)** |
|---|---|---|---|---|
| Population | 168,048 | 100.00 | 9237 | 100.00 |
| Age group 0–14 years | 21,860 | 13.01 | 1552 | 16.80 |
| Age group 65+ years | 27,258 | 16.22 | 833 | 9.02 |
| Tertiary education | 36,551 | 21.75 | 2242 | 24.27 |
| Active earners | 61,699 | 36.72 | 3917 | 42.41 |
| ISCO1–2 | 20,965 | 12.48 | 1381 | 14.95 |
| ISCO7–9 | 18,247 | 10.86 | 973 | 10.53 |
| Housing | 70,821 | 100.00 | 4018 | 100.00 |
| One-bedroom dwellings | 5988 | 8.46 | 569 | 14.16 |
| Four+−bedroom dwellings | 13,701 | 19.35 | 653 | 16.25 |
| Built before 1946 | 8919 | 12.59 | 487 | 12.12 |
| Built after 1990 | 13,425 | 18.96 | 2043 | 50.85 |
| Low-comfort dwellings | 1876 | 2.65 | 119 | 2.96 |

Source: Hungarian Central Statistical Office (HCSO), National Census 2011.

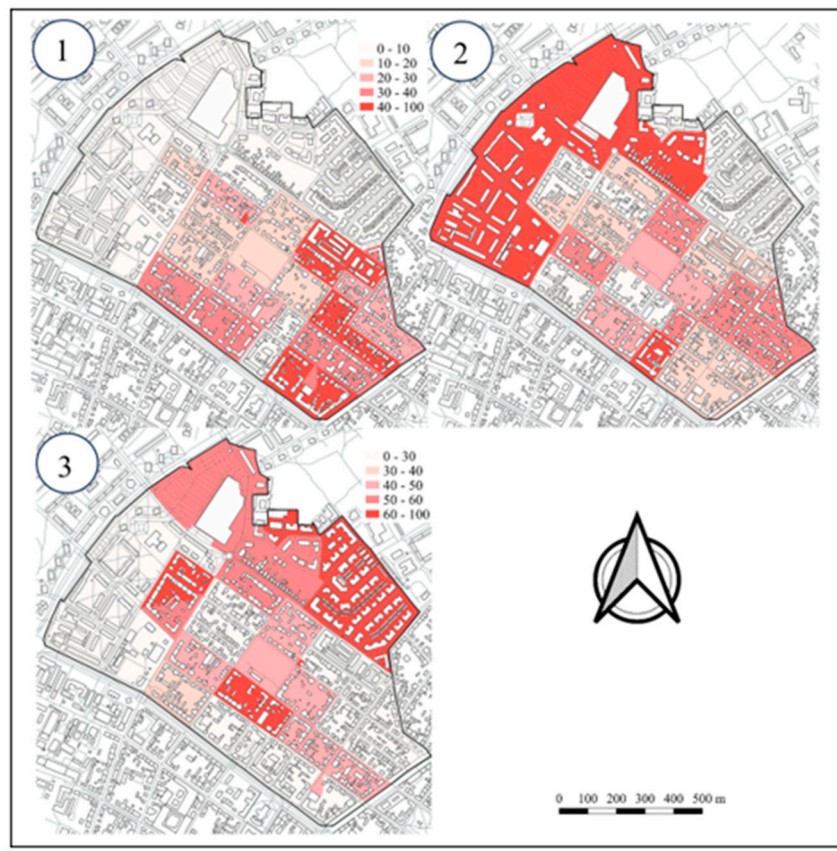

**Figure 2.** Share of housing built before 1946 (1), between 1946 and 1990 (2), and after 1990 (3) in Rókus (Szeged), Hungary.

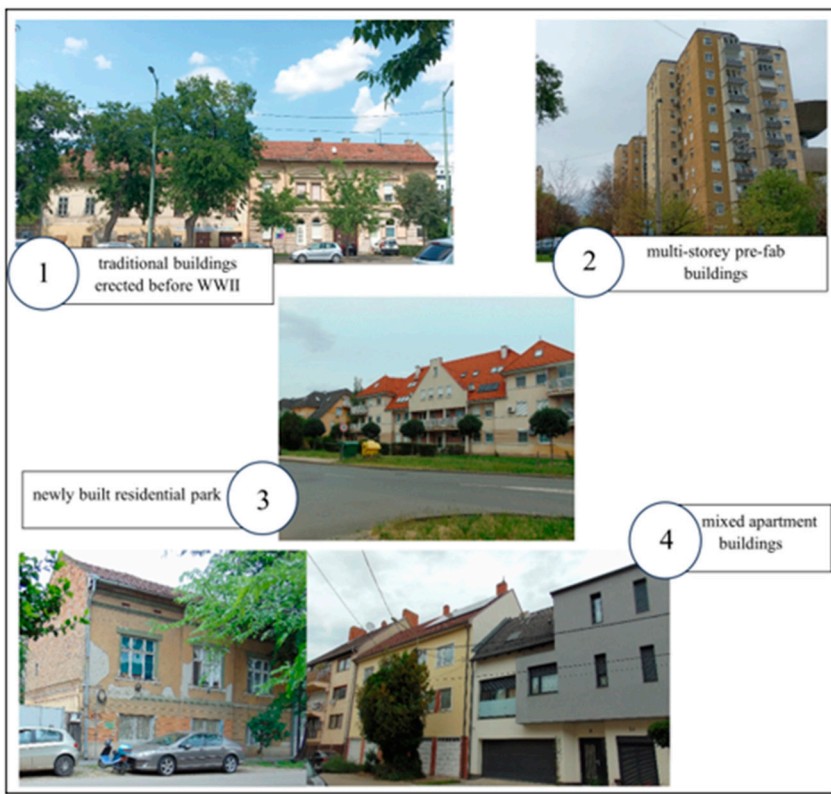

**Figure 3.** Different built-up forms in Rókus (Szeged), Hungary. Source: Hungarian Central Statistical Office, National Census 2011.

Between the two edges of the neighborhood, a transitional zone can be identified where the relative density of new, post-socialist housing, built after 1990 is high. This transitional zone where old (pre-WWII) and new (post-1990) housing are intermixed consists of two parts, namely the northeastern tracts, where the Franciahögy residential park is located, and the southern tracts, where old tenement blocks are mixed with new, upmarket infills. This type of development is also well-documented in other post-socialist countries [47,58]. Franciahögy was the first gated residential complex to be constructed in Szeged between 2000 and 2007 when this form of housing started to mushroom in Hungary at the turn of the new millennium [59]. It was developed with relatively dense, uniform, and monotonous apartment buildings, adhering to building regulations by making intensive use of the rooftops (see (3) of Figure 3). Thus, despite the buildings appearing to be 1–2 stories in height, there are two additional levels of living space underneath the tall and complex roof structures. Franciahögy is well-known both in the public consciousness and the real estate market in Szeged, and it is generally identified as a high-status, new housing development built for middle-class families. In the south of Franciahögy, in the center of Rókus, we find heterogeneous housing stock where pre-WWII and post-2000 apartment houses are mixed. New housing construction in the past two decades took place in the form of upmarket infills on vacant plots and as a replacement for old housing after demolition (see (4) of Figure 3).

Figure 4 shows that socio-spatial inequality within the study area is relatively high. Members of the top occupational categories (Managers and Professionals) tend to be concentrated in the southeastern part of the neighborhood (see Figure 4A), while lower socio-economic groups (laborers in industry, construction workers, machine operators, and other "elementary occupations") are overrepresented in the northwestern part of Rókus, where the housing market is dominated by multistorey prefabricated buildings (see Figure 4B). We were interested not only in the spatial segregation of the top and

bottom occupational categories but also in the level of diversity and polarization within census tracts.

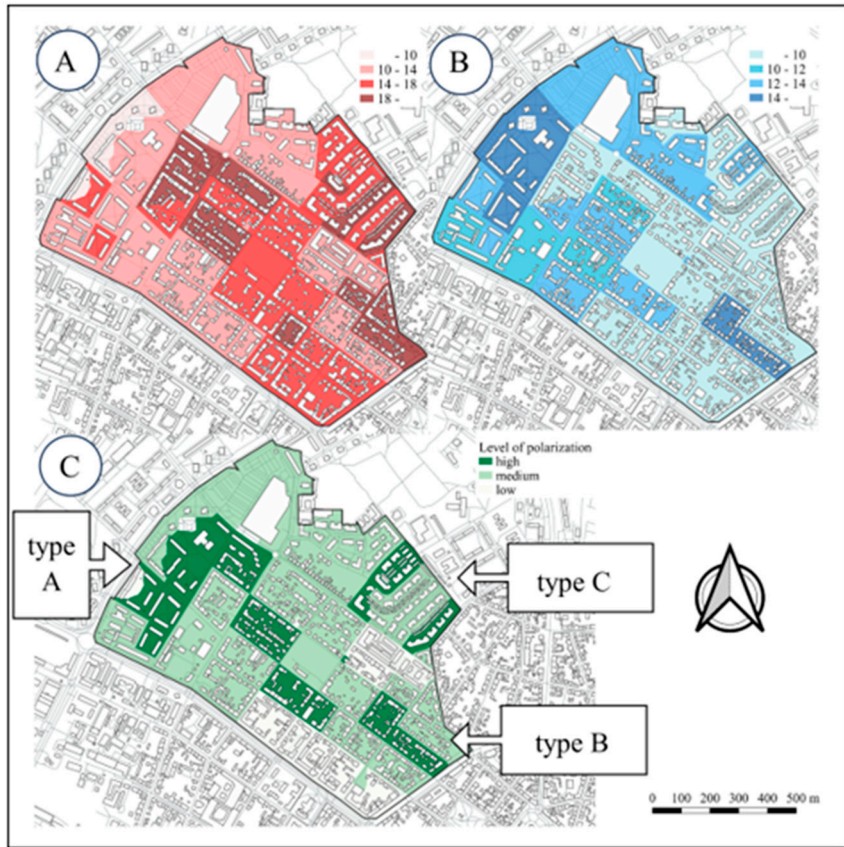

**Figure 4.** Share of top (**A**) and bottom (**B**) occupational groups, and the level of social polarization (**C**) in census tracts in Rókus (Szeged), Hungary. Source: Hungarian Central Statistical Office, National Census 2011.

As a result of diversity calculations, three distinct (see Figure 4C) areas could be identified in the sampled area. For each census tract, we examined whether there is a coincidence of social diversity and polarization, i.e., the above-average presence of the top (ISCO 1–2) and bottom (ISCO 7–9) occupational categories. Research results showed that diversity and polarization go hand in hand only partially. With a high level of diversity, in some cases, we also experienced lower levels of polarization between the ISCO groups. However, we found three distinct areas that were not only diversified but also polarized with higher shares of the top and bottom ISCO categories. These census tracts were depicted on our map in dark green.

The first type (A) is associated with prefabricated high-rise housing estates from the state-socialist period. As a result of the socialist housing allocation policies aimed at equalization, both top and bottom ISCO groups are concentrated with above-average shares in the area. In addition to easing housing shortages, the main aim of the development of large housing estates was to reduce spatial segregation within the society, allowing working-class and white-collar workers to mix in the newly erected housing estates. The results of these efforts are still noticeable today. Notably, prefabricated housing estates in general provide a good entry point for young families to the housing market as such smaller apartments can be obtained at an affordable price level. This, in turn, leads to the mixing of different social groups from both the top and bottom ISCO categories.

The second type (B) is represented by the transforming and dynamically changing area with a mix of historical civic houses and infill developments. In census tracts with infill-type developments, it is common to find two very different types of buildings (old

and new) on the same street. Furthermore, these stark contrasts regarding the age, quality, and aesthetic appearance of buildings may be observed next to each other. One building may be well-maintained, modern, and visually pleasing, while the next may be neglected or dilapidated. New buildings are integrated into the existing, already heterogeneous environment where different architectural styles and building qualities intersect. Such microscale disparities have consequences for the everyday life of residents, the level of interactions between them, power relations, and social cohesion.

The third type (Type C) is related to the Franciahögy residential park, which was built as an upmarket residential development some two decades ago. Here, a peculiar mix of horizontal and vertical segregation can be observed at the microscale due to the type of construction. According to real estate experts, apartments on lower levels are bigger and relatively more expensive, while attic conversions are smaller and less desirable due to the risk of leakage, stairs, and a smaller usable floor area. This creates vertical segregation within the buildings. In addition, buildings located on the edges of the residential park experience disadvantages as they are more exposed to traffic noise and vibration, resulting in lower property values. This type of pattern leads to both vertical and horizontal segregation in Franciahögy on the microscale.

### 4.2. Qualitative Aspects of Segregation in Rókus

The main aim of the in-depth interviews was to capture the invisible boundaries of social segregation, how they are perceived by the residents, and how they are constructed. Concerning the visible differences manifested in the urban fabric (described above), we were interested in how interviewees perceived the differences observed in the built environment of the neighborhood. According to the interviewees, in terms of physical appearance, the socio-spatial segregation of different areas in the multifaceted district is not so evident. The most distinct profile is attached to the Franciahögy residential park, and according to the interviewees, the residential park stands out from the rest of Rókus in both architectural and social aspects. In the residential park of Franciahögy, it is noticeable that families move in because of the small green gardens and playground [which reflect their lifestyle-driven preferences for real estate], says one interviewee. As mentioned earlier, the Franciahögy development differs from other parts of Rókus both in terms of the real estate market and social indicators, and this difference is confirmed by the interviewees. According to one high school teacher, it is becoming increasingly common for parents/students to choose a school near their place of residence. Many students come from nearby high-quality schools, thanks to the residential park in the area. More young families are living in the park, and they send their children to this school. Regarding the general perception of schools, a teacher at a low-status technical school observes that few students from the nearby residential park come to the technical/vocational school, whereas significantly more students come from the housing estate environment to the institution. This indicates that residents of the residential park, who are wealthier, tend to choose the higher-ranked grammar school, which creates segregation invisibly in the area.

In other parts of the neighborhood, which differ in physical character (prefabricated high-rise buildings and single-family houses), there is no perceptible socio-spatial segregation according to the respondents. Franciahögy is considered a divisive area among buyers due to the quality of the buildings and its previous function (cemetery), says a real estate interviewee; however, there is no strong opinion regarding the other areas. Although the study area is characterized by more valuable and less valuable parts from a property market perspective and streets to be avoided, the sense of a diverse community and the diverse composition of the population is more prevalent, which is also reflected in the real estate market value. The third part [mixed residential part of Rókus] is completely mixed, and the buyers cannot be delineated by a single type. One interviewee explains that the transforming area, between Hétvezér Street and Tavasz Street, is not among the popular destinations, but there are differences within the streets, says another interviewee. Therefore, in the case of Rókus, socio-spatial segregation is based not on larger spatial units,

such as census tracts or blocks, but rather on some smaller microscale entities (e.g., specific streets or territorial clusters). One interviewee also points out that generally, the higher the share of people living in municipal rental housing (IKV) within the block, the worse it is, indicating that housing tenure and the related socio-economic conditions also influence the perception of the area. The different merits and people's perceptions of various parts of the study area became evident from the interviews. Although some respondents generally find the neighborhood beautiful, livable, and safe, there is a growing concern regarding the area being noisy when closer to the main road.

The sense of segregation in the neighborhood is more influenced by the visible factors (appearance), and condition of the built environment rather than the other way around. This inverse relationship is reflected in the responses of the interviewees as they believe that socially lower-status households tend to live in low-quality and older buildings. According to one real estate agent, the study area can be divided into separate subareas with different buyer profiles. The vicinity of the boulevard, the mixed-use area, and the panel buildings can all be considered more affordable housing, the former because it is located slightly farther from the city center and the latter due to its general perception.

As the results of the interviews showed, there is no significant segregation based on ethnicity in the Rókus neighborhood. The respondents mentioned the presence of Roma people in the area, but there is no noticeable contrast. The interviewees consider the neighborhood to be safe, and they do not have a sense of fear. There are certain streets where Roma people are more prevalent; however, the respondents believe that it is possible to avoid those areas. Some interviewees mentioned that people in the neighborhood are divisive, indicating a diverse population. One respondent who lives in the mixed part of the neighborhood appreciates the sense of inclusion brought about by the mixing of social groups: I believe it prevents anyone from being marginalized, including the homeless and the poor. When asked about conflicts arising from ethnic differences, only one respondent reported a verbal conflict involving Roma people, but they do not consider the area dangerous or prone to segregation. They mentioned that the Roma people were shouting and begging on the street. Some interviewees mentioned that the area is affected by the presence of homeless people and individuals struggling with alcoholism, both men and women are among them. In addition, there is a person with mental health issues residing in one of the panel buildings, who has been threatening the residents. Unfortunately, the residents have not received any assistance in resolving this situation. The presence of a mixed population is demonstrated by an interviewee stating that neighbors were harassing each other, which is a sign of conflicts related to land use.

A primary school teacher mentioned that the local primary school has students from different (lower) social groups every year, but their number is small, and their peers are accepting of them. Another interviewee stated that the lack of a secure social background and stable family represents segregation within schools. This underpins that the multiple family factor is the reason for potential segregation in schools, and it is not only the ethnicity or the color of skin. Even though people perceive no strong segregation, some interviewees mentioned that "the Roma community tends to keep to themselves away [from the majority when using public spaces]", which could be seen as a form of mild segregation. They also noted that there is no real interaction with the Roma, and therefore, their presence in the neighborhood was not causing any problems. The income situation in the neighborhood is indicated by an interviewee who mentioned, "There are few truly impoverished individuals [...but] income does not determine the dynamics of the student community". Another interviewee did not emphasize ethnic divisions, but rather, highlighted differences caused by wealth disparities. She stated, "Roma children do not cause a problem in school; they perform well, but there is an increasing number of young influential parents [...] segregation becomes noticeable not only among parents but also among children; snobbishness tends to emerge". Some of the interviewees perceive segregation and "cliques" among both the wealthier and the poorer groups; however, at the end of the day, most respondents

have a good relationship with their neighbors and actively initiate conversations within the neighborhood.

To assess possible forms of social segregation manifested in lifestyle, we asked interviewees for reflections regarding their leisure activities, hobbies, and living conditions. Based on the responses, regardless of their workplace position and occupational status, there were no significant differences in how they spent their free time. Furthermore, the diverse hobbies reflect, to some extent, their income status. The most common hobbies mentioned include reading, sports (such as tennis, home workouts, team sports, cycling, and Pilates), hiking, and pottery making. In addition to these pursuits, interviewees also expressed concern about the lack of community spaces and playgrounds, which limits the outdoor activities in the neighborhood.

According to the statistical analysis, the study area has no highly segregated parts and is not endangered by severe forms of segregation. This is also influenced by the territorial division of census tracts. The territorial units are established at the local level, and they divide the problematic streets, Tavasz and Hétvezér. Therefore, the higher proportion of the Roma population might not be reflected in the area due to the way the census units were defined. As confirmed by one real estate agent interviewee, it is observed in every neighborhood in Szeged, including Rókus, that there is at least one less safe street, and if you go one street further or around the corner, it is already considered a good neighborhood. One interviewee highlights the process of filtration and a small-scale concentration of Roma people. "[Roma population] tends to cluster in smaller focal points within the city, such as certain houses or blocks. Houses where only Roma people lived were considered more turbulent areas [. . .]; however, these buildings were demolished, and new ones were built in their place".

Based on the experiences from field visits and interviews conducted with real estate experts and residents, the changing character of the area is also sustained by population change. It is often observed that in the vicinity of older and poorly maintained buildings, new multi-apartment condominiums are being constructed as infill developments. As a result, within the area, the sense of a diverse social community mentioned earlier is formed among the locals. In addition, the social polarization of the neighborhood is visibly and partially perceptible as a parallel process. The condition and character of the buildings influence the perception of the property and its surroundings. One real estate agent mentions several factors that contribute to the lower value of old, historical civic houses and newly built attic apartments. A ground-floor or a first-floor property is not as bright as a top-floor one, as they lack sufficient sunlight, making them more challenging to sell and only at a lower price. Concerning top-floor apartments, many people fear that the roof might leak, which also devalues and lowers their prestige in the eyes of buyers. All these affect the target market as well.

## 5. Discussion and Conclusions

This study aimed to investigate existing forms of microsegregation in a selected neighborhood of Szeged (Hungary) with a complex methodology. Socio-economic data from the last census (2011) revealed clear differences between the census tracts regarding the level of education, occupational status, and housing conditions of the residents. The Franciahögy residential park is a very distinct and highly segregated area within the neighborhood, which was confirmed both by statistics and interviews. However, our results also showed that even within this microsegment of the study area, further manifestations of microsegregation exist both in horizontal and vertical terms. Outside Franciahögy, the study area can be considered rather heterogeneous but less segregated.

The findings highlighted the interplay of diversity and polarization in two different ways. Higher diversity generally reduces the likelihood of polarization; however, when the presence of two opposite groups (top and bottom ISCO categories) is the source of diversity, the level of polarization is also high. In the sample area, we identified three distinct areas that fall in the latter category. They represent three distinct types of urban

fabric that are the results of past and present housing policies. One of them is connected to the prefabricated high-rise estates from state socialism, another represents the postmodern gated-community-like built-up area, and the third is manifested by infill developments embedded in a transitional area. The reasons behind the higher potential for polarization are multifold in all cases; nevertheless, all three provide good examples of microsegregation in both horizontal and vertical terms. One of the main findings of our research is that different forms of microsegregation can coexist even in one relatively small neighborhood due to the intersectionality of different factors; however, common in all three cases is that the sorting effect of the housing market is prevalent.

The boundaries of segregation are constructed through the interplay of visible and invisible elements, as demonstrated by the statistical analysis and qualitative research findings (Figure 5). The visible factors contributing to segregation include the distinct and unique building stock of different blocks, urban fabric, and the overall character of the neighborhood. The boundaries of perceived segregation within the neighborhood can be attributed to several factors. First, the neighborhood has distinct internal boundaries as far as the built-up areas and architectural dividing lines are concerned. The most distinct boundaries are related to the post-socialist residential park (Franciahögy) and the socialist prefabricated housing estate. The interviewees confirmed the diversity of the neighborhood, but instead of highlighting the ethnic aspect of segregation, the common interest development residential complex, Franciahögy, was mentioned as a higher-class enclave, which also produces clear-cut educational segregation.

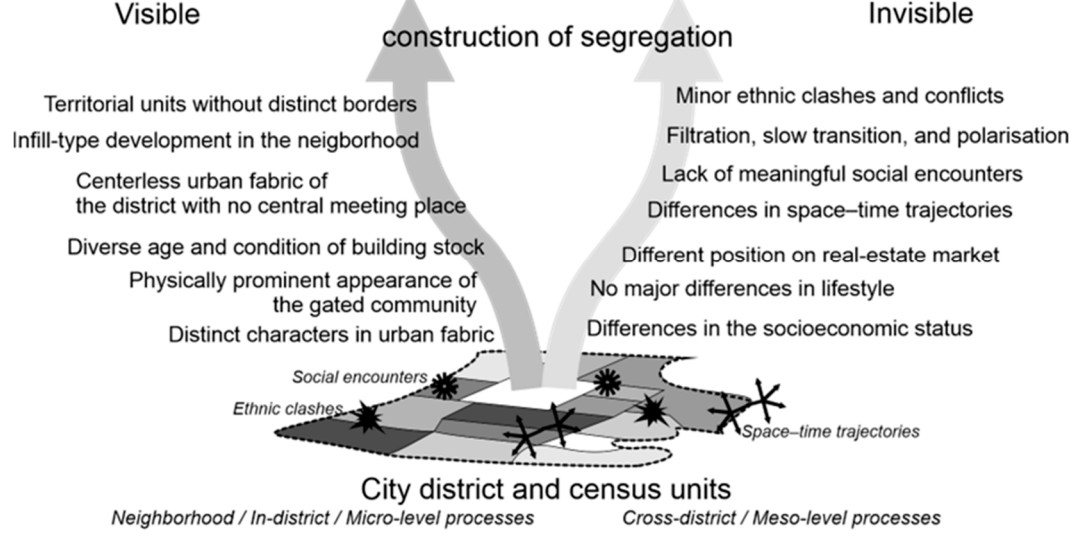

**Figure 5.** The making of boundaries of microsegregation.

These visible boundaries play a crucial role in the mental maps of individuals regarding segregation. In their absence, it becomes difficult to delineate areas that are distinct from others based on different physical and social factors. Moreover, it is important to consider that the boundaries between these microspaces are not always clearly defined but, rather, diffuse with areas often transitioning into one another. In addition to the absence of clear boundaries, the study area has undergone a dynamic transformation in recent decades, i.e., a step-by-step upgrade thanks to numerous infill-type developments. Being in the transitional zone of the city, the local building regulations do not impose strict requirements on the built-up character. The gradual renewal and transformation of the building stock of the neighborhood hinders the delineation of distinct areas. Moreover, since there are no larger units with distinct social and architectural characteristics apart from the Franciahögy residential park and the prefabricated housing estate, identifying boundaries of microsegregation is also a challenge for the residents.

The lack of a sense of segregation has been reinforced by the interviewees in that in the neighborhood it is not possible to perceive differences partly due to the lack of public spaces where daily encounters between people from different social groups could take place. The main square in the middle of the neighborhood comprises a soccer field with a small playground, serving very specific functions, and their use varies greatly among different social groups over time. The lack of encounters prevents everyday interpersonal and intergroup interactions that could provide a basis for negative stereotypes and more pronounced segregation [60]. Therefore, the socio-spatial differences demonstrated by statistical indicators remain invisible due to the absence of encounters.

The lack of infrastructure for meaningful social encounters highlights that it is equally important to recognize the invisible factors at play in segregation. These invisible factors include social dynamics, historical contexts, socio-economic disparities, systemic inequalities, and mental and cognitive perceptions of the residents. According to our findings, wealth disparities linked with occupation are the main source of microsegregation. Such invisible factors often remain unnoticed or hidden; however, sometimes they are tied to the visible (i.e., built-up) elements. The infill redevelopment leads to a displacement process where younger and better-off households are gradually becoming overrepresented in certain blocks and streets, while people of long-term lower status are disappearing. The increasing number of young and influential parents in the neighborhood contributes to the segregation of children in school, paralleled by the emergence of snobbishness among both parents and children. In addition, the perceived differences in property values also contribute to the invisible segregation process. Although ethnicity does not play a significant role in the life of the local community, some minor ethnic tensions (between the Roma and Hungarians) contribute to a conscious segregation process, influencing the space–time trajectories of residents and shaping the local pattern of microsegregation. This brings us to the topic of social justice—a key principle underpinning much work in contemporary social geography [61,62]. Our study on the lived experiences of social inequalities and microsegregation demonstrates how tensions between social justice and everyday city life can arise and provides an opportunity to consider what policies might be put in place to make social justice a reality. In this respect, we propose three considerations for policymakers.

1. The concept of segregation at the local level should be redefined. In the study area, segregation processes do not manifest statistically because Hungarian legislation treats segregation as a fundamentally ethnic-based process. This approach should be changed and other factors such as income, educational level, occupation, age, gender, and other factors of segregation should be considered in local planning and policy documents.

2. Part of the observed microsegregation processes in the neighborhood can be attributed to the lack of social connections and encounters. To change this situation, communal spaces within the neighborhood that provide encounters and places for interactions between diverse socio-economic groups should be developed. In addition, community programs and cultural events organized within the neighborhood could facilitate interactions and coexistence between various societal groups and enhance social cohesion. Involving local communities in decision-making processes and dialogue can also contribute to collaboration and understanding between different groups.

3. A further challenging issue in the neighborhood under study is the resurgence of lifestyle- and wealth-based segregation processes in educational institutions. Initiatives such as awareness campaigns and educational programs emphasizing social equality could help reduce prejudices and promote acceptance among young people.

Our research findings suggest that while the overall perception of the neighborhood is good, without greater tensions, there are obvious socio-spatial differences and signs of segregation at the microscale. The daily experiences and perceptions of respondents shed light on these nuances and indicate that while the area may appear cohesive, several factors contribute to socio-economic and cultural disparities in the area and the emergence of the boundaries of segregation. Moreover, our study confirmed the importance of considering

both the macrolevel uniformity and microlevel differentiations when discussing segregation in a particular neighborhood. Understanding segregation requires a multiscalar and interdisciplinary approach that combines statistical analysis with qualitative insights. This holistic perspective may help to shed light on the complex and nuanced nature of segregation, allowing further academic debate on microscale segregation processes.

**Author Contributions:** Conceptualization, Z.K. and G.N.; methodology, G.N. and R.V.; formal analysis, R.V., G.N. and Z.K.; investigation, R.V. and G.N.; resources, Z.K.; writing—original draft preparation, R.V., G.N. and Z.K.; writing—review and editing, Z.K. and G.N.; visualization, R.V. and G.N.; supervision, Z.K.; project administration, Z.K.; funding acquisition, Z.K. All authors have read and agreed to the published version of the manuscript.

**Funding:** This research has been funded by the Hungarian Scientific Research Fund (OTKA) Grant Agreement No. K135546, and the National Research, Development and Innovation Fund of the Ministry of Innovation and Technology, Hungary, grant number TKP2021-NVA-09, and the Ministry for Culture and Innovation, through the National Research, Development and Innovation Office, using the funds earmarked from the National Research, Development and Innovation Fund within the framework of the New National Excellence Program grant No. ÚNKP-23-3–SZTE–530.

**Conflicts of Interest:** The authors declare no conflict of interest. The funders had no role in the design of the study; in the collection, analyses, or interpretation of data; in the writing of the manuscript, or in the decision to publish the results.

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
