# Peer review of "The Construction of the Visible and Invisible Boundaries of Microsegregation: A Case Study from Szeged, Hungary"

_land, doi:10.3390/land12101835_

Round 1
Reviewer 1 Report
Comment 1: The introduction does not sufficiently problematizes the problem investigated and is limited to presenting the structure of the manuscript;
Comment 2: The english style should be more polite. There are many excerpts that could be better written and there is no cohesion and textual consistency;
Comment 3: As this is a research article, it is not pertinent that there is such a literature review. This could be incorporated throughout the discussions;
Comment 4: The location map is not clear enough where the study area is (in Europe) and the cartographic representation (map caption);
Comment 5: There is no clarity as to the objective of the research. There is also no structure of methodological procedures that endure with robustness and rigor to obtain results;
Comment 6: The discussions and conclusions are in the same section. There is no specific separation between these components. There is also no relationship with other research on the same theme under similar conditions.
The english style should be more polite.
Author Response
Response to Comment 1: Thank you for this (and also another reviewer’s) comment. The introduction of the manuscript was extended, the research objective is now more clearly articulated, and a new research approach (intersectionality) is emphasized, with relevant citations.
Response to Comment 2: The text was proofread again and we tried to improve the consistency and grammar of the text as much as it was possible.
Response to Comment 3: Given the versatile theoretical foundation of the research and its wider methodological relevance we thought a section portraying the most important advancements and related literature in the field was needed. Other reviewers asked even the extension of the state-of-the-art, what we had to take into consideration.
Response to Comment 4: Figure 1 was redesigned accordingly, now the new figure positions the study area within Europe.
Response to Comment 5: The objective of the research is now better elaborated and clarified. The methodological section is very detailed it gives sufficient information on how the quantitative data analysis and the qualitative research were performed. The complex research method provides a solid basis for the robustness and novelty of the research findings.
Response to Comment 6: The connection with other research in the field is articulated now in the introduction and the literature review sections. New references were included in the conclusion section on the relevance of the research findings with regard to social justice and everyday city life. In addition, some policy recommendations are formulated based on the results.
During the revision a language editing was performed.
Reviewer 2 Report
This manuscript used quantitative and qualitative methods to identify the visible and invisible boundaries of microsegregation in one of the neighbourhoods in Szeged, Hungary. The methodology design is appropriate, and the representation of results is clear. I recommend publication after minor revision.
1. Line 110-113. Their?
2. Line 223-225. Why do the authors think it was sufficient to use 2011 census data? Please give explanations in the context.
3. Line 274-275. Semi-structured or structured?
4. Please specify the building types in Figure 3, and legends in Figure 2 and Figure 4.
5. Any policy/future research recommendations to minimise the visible and invisible boundaries?
It is easy and clear to follow the whole context.
Author Response
Thank you for your overall evaluation.
- Sorry for this mistake. We have corrected it, and the text now flows: “Most recently, in their study on Hong Kong Ho and Yip [29] introduced even the concept of nano-segregation…”
- Even though the last census of Hungary was held in November 2022, but data on the census tract level are available yet. Therefore, we used 2011 census data, to unpack micro-scale socio-economic inequalities within the case-study area. Since Hungary and the city of Szeged has an aging society, where changes in the demographic and social structure of the local population are occurring slowly, and the intensity of residential mobility is also low, we thought as proxy indicators the 2011 census data can well indicate the internal socio-economic division of the neighborhood.
- The answer is clear: semi-structured
- The caption of Figure 2 was changed; letters were replaced by numbers. The legend of Figure 3 was changed with text that clarifies the building types. The content of Figure 4 is clear in the light of the text.
- The conclusions were extended by some policy recommendations.
Reviewer 3 Report
Thank you for the opportunity to read and review such an interesting manuscript. My expertise is primarily in feminist and social geographies, social theory, as well as qualitative methodologies - therefore, I will not comment much on how the paper relates to Hungarian geographies, because I am less familiar with that regional literature. In my estimation, the paper is a significant original contribution to urban social geography. The writing is of decent quality, and the content is fresh and interesting, which makes for a wonderful reading experience. The paper has obvious practical applicability, furthering the recent call in critical geography to get outside the Ivory tower of academia and to contribute more directly to making the world a better place, through activism and/or relevant technical expertise. Given the broad relevance of the paper, I think that there is an excellent fit with the aims and scope of Land. Given all these strengths, I am inclined to recommend publication of the manuscript in this journal, but before being able to do so, I would need to see a revised version of the paper that addresses a couple of current weaknesses, as follows:
(1) The paper under-theorizes the concept of local geographical context, despite its centrality to the argument. The author should at least mention if not discuss the recent theorizing of this problematic by juxtaposing personal change and social change, and how the latter constitutes a spatio-temporal context for the former [Key reference: Simandan, D., 2020. Being surprised and surprising ourselves: a geography of personal and social change. Progress in Human Geography, 44(1), pp.99-118; another useful reference for this is Drozdzewski, D. and Webster, N.A., 2021. (Re) visiting the neighbourhood. Geography Compass, 15(12), p.e12597.]. This would require perhaps two or three sentences to be added to the literature review.
(2) The authors see the paper as contributing to social geography, but contemporary social geography has been significantly influenced by feminist theorizing of concepts of intersectionality, positionality, and situated knowledge. So it is striking that the authors do not any attempt to situate themselves epistemologically by accounting for their positionality. I think the introduction of the paper needs an addition of a couple of sentences that more directly reference and speak to the recent literature on these feminist/social geography problematics (the two key references I think are worth mentioning are [Hopkins, P., 2019. Social geography I: intersectionality. Progress in Human Geography, 43(5), pp.937-947.] and [Simandan, D., 2019. Revisiting positionality and the thesis of situated knowledge. Dialogues in human geography, 9(2), pp.129-149.]). In other words, by anchoring the paper in this literature, the authors would more explicitly account for the situatedness of their knowledge claims, and for promoting a type of social geography writing that values an intersectional and situated sensibility.
(3) The last minor correction I would like to see is about the centrality of social justice to contemporary social geography. By highlighting this centrality in the conclusion of the paper, the author would better articulate the tensions between social justice and everyday city life (in Szeged), and what social geography has to contribute to foregrounding those tensions, for both analytical and political purposes. The reference that I think is most useful for developing this point is Hopkins, P., 2020. Social geography III: Committing to social justice. Progress in Human Geography, https://journals.sagepub.com/doi/full/10.1177/0309132520913612, but Clive Barnett's work might also be worth mentioning [Barnett, C., 2018. Geography and the priority of injustice. Annals of the American Association of Geographers, 108(2), pp.317-326.]. I don't think this requires altering the existing layout - I have in mind just a minor addition of a couple of sentences to the conclusion.
Minor editing required.
Author Response
Thank you for the general evaluation of our manuscript. During the revision, we tried to solve all the shortcomings, address all the weaknesses, and consider all your valuable recommendations.
(1) Thank you for this comment and the recommended literature. In fact, we could make good use of the suggested papers and also others we found in the field, in extending the literature review. Following Simandan’s work we put a special emphasis on the transformation of space and society across geographical scales, juxtaposing personal change and wider societal change. We also tried to emphasize the importance of understanding of the neighborhood in segregation studies quoting the work of Drozdzewski & Webster (2021), and underlying that the neighborhood is a multi-scalar spatial unit, that should always be considered by social geographers.
(2) We are grateful for your comments and the recommended papers. The introduction of the manuscript was extended with direct reference to intersectionality, citing the most relevant literature, and emphasizing that intersectionality is a useful tool for mapping and understanding inequalities in urban space.
(3) Thank you for this comment, in the conclusion of the paper we clearly state the tensions between social justice and everyday city life, citing the recommended papers and also providing some policy recommendations.
During the revision, some minor language editing was performed.
Round 2
Reviewer 1 Report
The authors made all the suggested improvements.
Congratulations!
Kind Regards.
Reviewer 3 Report
Thank you for engaging so thoroughly with the reviewer feedback. I have no further corrections required.
N/A